# First Report of *aacC5-aadA7Δ4* Gene Cassette Array and Phage Tail Tape Measure Protein on Class 1 Integrons of *Campylobacter* Species Isolated from Animal and Human Sources in Egypt

**DOI:** 10.3390/ani10112067

**Published:** 2020-11-08

**Authors:** Norhan K. Abd El-Aziz, Ahmed M. Ammar, Mona M. Hamdy, Adil A. Gobouri, Ehab Azab, Alaa H. Sewid

**Affiliations:** 1Department of Microbiology, Faculty of Veterinary Medicine, Zagazig University, Zagazig 44511, Egypt; prof.ahmedammar_2000@yahoo.com (A.M.A.); dr.mona_micro@yahoo.com (M.M.H.); veterinarian.alaa.sweed@gmail.com (A.H.S.); 2Department of Chemistry, College of Science, Taif University, P.O. Box 11099, Taif 21944, Saudi Arabia; a.gobouri@tu.edu.sa; 3Department of Biotechnology, College of Science, Taif University, P.O. Box 11099, Taif 21944, Saudi Arabia; e.azab@tu.edu.sa

**Keywords:** *Campylobacter* species, class 1 integrons, extensively drug-resistance, pan drug-resistance, gene cassette arrays

## Abstract

**Simple Summary:**

*Campylobacter* species are among the major causes of bacterial foodborne infections. Here, we investigate, for the first time, class 1 integrons and associated gene cassettes among pan drug-resistant (PDR), extensively drug-resistant (XDR), and multidrug-resistant (MDR) *Campylobacter* species isolated from livestock animals and humans in Egypt. Our results revealed alarming PDR (2.55%) and inordinate XDR (68.94%) and MDR (28.5%) *Campylobacter* isolates. None of the examined isolates were pan-susceptible. The existence of a novel gene cassette array, namely *aacC5-aadA7Δ4* and a putative phage tail tape measure protein on class 1 integrons of *Campylobacter* species is the most highlighted novelty of the current study. Evidence from this study showed the possibility of *Campylobacter*–bacteriophage interactions as well as treatment failure in animals and humans due to horizontal gene transfer mediated by class 1 integrons.

**Abstract:**

*Campylobacter* species are common commensals in the gastrointestinal tract of livestock animals; thus, animal-to-human transmission occurs frequently. We investigated for the first time, class 1 integrons and associated gene cassettes among pan drug-resistant (PDR), extensively drug-resistant (XDR), and multidrug-resistant (MDR) *Campylobacter* species isolated from livestock animals and humans in Egypt. *Campylobacter* species were detected in 58.11% of the analyzed chicken samples represented as 67.53% *Campylobacter jejuni*
*(C. jejuni*) and 32.47% *Campylobacter coli* (*C. coli*). *C. jejuni* isolates were reported in 51.42%, 74.28%, and 66.67% of examined minced meat, raw milk, and human stool samples, respectively. Variable antimicrobial resistance phenotypes; PDR (2.55%), XDR (68.94%), and MDR (28.5%) campylobacters were reported. Molecular analysis revealed that 97.36% of examined campylobacters were *integrase* gene-positive; all harbored the class 1 integrons, except one possessed an empty integron structure. DNA sequence analysis revealed the predominance of *aadA* (81.08%) and *dfrA* (67.56%) alleles accounting for resistance to aminoglycosides and trimethoprim, respectively. This is the first report of *aacC5-aadA7Δ4* gene cassette array and a putative phage tail tape measure protein on class 1 integrons of *Campylobacter* isolates. Evidence from this study showed the possibility of *Campylobacter*–bacteriophage interactions and treatment failure in animals and humans due to horizontal gene transfer mediated by class 1 integrons.

## 1. Introduction

Thermophilic *Campylobacter* species, particularly *Campylobacter jejuni* (*C. jejuni*) and *Campylobacter coli* (*C. coli*) pose veterinary and public health concerns due to their zoonotic potential, the enormous range of reservoir hosts, and persistence in the environment [1]. Consumption of contaminated food, especially poultry products, unpasteurized milk, and undercooked meat, as well as water, is a risk factor for *C. jejuni* and *C. coli* infections [2,3].

Most *Campylobacter* infections are self-limiting and require no therapeutic intervention other than supportive and rehydration therapy. However, prompt antimicrobial treatment is employed in immunocompromised individuals, patients whose symptoms are severe or persistent, intense or prolonged enteritis, cases of bacteremia, and those with extraintestinal infections [4]. Attention to the resistance of campylobacters has been launched due to the indiscriminate abuse of antibiotics [5]. Clinical, veterinary, and environmental surveys have shown that bacteria harboring integrons are frequently associated with the multidrug-resistant (MDR) phenotype [6]. However, the extensively drug-resistant (XDR) and pan drug-resistant (PDR) bacteria are epidemiologically significant not only due to their resistance to multiple antimicrobial agents but also to their ominous prospect of being resistant to almost all or all approved antimicrobial agents [7,8].

The integron is a site-specific recombination system capable of integrating mobile gene cassettes, which can be expressed and disseminated via horizontal gene transfer [9,10]. Class 1 integron includes two conserved segments (CSs), denoted as 5′- and 3′-CSs, flanking a gene cassette. An *int1* gene encoding an integrase enzyme is located within the 5′-CS and is responsible for the recombination of a gene cassette [11]. The 3′-CS possesses *qacE∆1* and *sul1* genes encoding resistance to quaternary ammonium compounds and sulfonamide, respectively. Integrons can incorporate and express more than one gene cassette conferring resistance to multiple antimicrobial classes such as beta-lactams, aminoglycosides, trimethoprim, chloramphenicol, fosfomycin, macrolides, lincosamides, rifampicin, and quinolones [11].

Previous studies reported class 1 integrons carried aminoglycoside (*aadA* and *aacA4*) and trimethoprim (*dfr1* and *dfr9*) resistance gene cassettes in both *C. jejuni* and *C. coli* isolated from chicken house environment [12], poultry, and human sources [13,14,15,16] without showing the antimicrobial resistance profiles of the isolates. However, these genetic elements were not detected in an XDR *C. jejuni* CCARM 13,322 isolate recovered from a human case of diarrhea associated with international travel [17]. Hence, this study reports, for the first time, class 1 integrons and associated gene cassettes in thermophilic *Campylobacter* species isolated from livestock animals and humans showing variable antimicrobial resistance phenotypes.

## 2. Materials and Methods

### 2.1. Samples

A total of 550 samples comprising chickens and chicken products (*n* = 265), meat and meat products (*n* = 160), milk and milk products (*n* = 95) as well as human stools (*n* = 30) were collected during the period from January 2018 to December 2019. Samples of animal origins were obtained from various retail outlets, Zagazig city, Sharkia Governorate, Egypt. Human samples were collected from patients being affected by diarrhea and gastroenteritis, attending various private laboratories located in Zagazig city. The samples were transported immediately in an icebox to the bacteriology laboratory for further analysis. The animal study was approved by the Animal Welfare and Research Ethics Committee, Faculty of Veterinary Medicine, Zagazig University. The human study was conducted following the Ethics of the World Medical Association (Declaration of Helsinki) and was approved by the research ethics committee of the Faculty of Medicine, Zagazig University (ApprovalNoZU-IRB#2056-18-05-2019). The patients participating in the research study provided written informed consent.

### 2.2. Bacteriological Analysis and Molecular Identification

Isolation of *Campylobacter* species was performed under microaerobic conditions according to the protocol established by Vandepitte et al. [18]. Samples were enriched in Preston *Campylobacter* selective enrichment broth (Oxoid, Cambridge, UK) at 42 °C for 48 h. The enrichment broth was plated onto modified charcoal cefoperazone deoxycholate agar (mCCDA; Oxoid, Cambridge, UK) then transferred onto Columbia agar (Oxoid, Cambridge, UK) plates supplemented with 5% sterile defibrinated horse blood. Presumptive *Campylobacter* colonies were confirmed by oxidase, catalase, hippurate, and indoxyl acetate hydrolyses biochemical tests, in addition to testing their susceptibilities to nalidixic acid and cephalothin antimicrobials (30 mg/disc, each) [19]. The bacterial DNA was extracted using a QIAamp DNA Mini kit (Qiagen GmbH, Hilden, Germany) according to the manufacturer’s instructions. Polymerase chain reaction (PCR) amplifications of the *23S rRNA* gene of *Campylobacter* species [20] in addition to *mapA* and *ceuE* genes of *C. jejuni* and *C. coli*, respectively, [21] were applied using oligonucleotide primers listed in Appendix A.

### 2.3. Antimicrobial Susceptibility Testing

The antimicrobial susceptibilities of *Campylobacter* isolates were tested on Mueller–Hinton agar media (Oxoid-CM0337B, Cambridge, UK) supplemented with 5% sterile defibrinated horse blood under microaerobic conditions using the disc diffusion method [22] following the guidelines of the Clinical and Laboratory Standards Institute (CLSI) [23]. A panel of 25 standard antimicrobial discs (Oxoid, Cambridge, UK) within different 14 antimicrobial categories were examined including penicillins [ampicillin (AM; 10 µg) and amoxicillin (AX; 25 µg)], penicillin combinations [ampicillin-sulbactam (SAM; 20/10 µg) and amoxycillin-clavulanic acid (AMC; 20/10 µg)], cephalosporines [cephalothin (KF; 30 µg), cefoxitin (FOX; 30 µg), cefoperazone (CEP; 75 µg) and cefepime (FEP; 30 µg)], carbapenemes [meropenem (MEM; 10 µg)], monobactams [azetronam (ATM; 30 µg)], aminoglycosides [streptomycin (S; 10 µg), tobramycin (TOB; 10 µg), gentamycin (CN; 10 µg) and amikacin (AK; 30 µg)], macrolides [erythromycin (E; 15 µg), azithromycin (AZM; 15 µg) and clarithromycin (CLR; 15 µg)], quinolones [nalidixic acid (NA; 30 µg) and ciprofloxacin (CIP; 5 µg)], sulfonamides [sulfamethoxazole-trimethoprim (SXT; 23.75/1.25 μg)], amphenicols [chloramphenicol (C; 30 µg)], polypeptides [colistin (CT; 10 µg)], oxazolidones [lenzolid (LNZ; 30 µg)], lincosamides [clindamycin (DA; 2 µg)] and tetracyclines [doxycycline (DO; 30 µg)]. The interpretive criteria of CLSI (for most antimicrobials) [23] or the European Committee for Antimicrobial Susceptibility Testing (EUCAST) (for macrolides) were followed to classify *Campylobacter* isolates as susceptible, intermediate, or resistant [24].

The multiple antimicrobial resistance (MAR) indices were calculated as previously reported [25]. Pan drug-resistance (resistance to all antimicrobial agents), extensive drug-resistance (resistance to all classes of antimicrobial agents except 2 or fewer), and multidrug-resistance (resistance to three or more classes of antimicrobial agents) were determined as reported elsewhere [26].

### 2.4. PCR Amplification of Class 1 Integrons and Associated Gene Cassettes

*Campylobacter* isolates exhibited variable antimicrobial resistance profiles (PDR, XDR, and MDR) were subjected to DNA extraction, using the QIAamp DNA Mini kit (Qiagen, Gmbh, Hilden, Germany) following the manufacturer’s recommendations. The isolates were screened for possession of the *integrase* gene as well as class 1 integrons using intI1 and hep primer sets, respectively [27,28] (Appendix A). Isolates containing class 1 integrons were screened for the existence of contiguous resistance gene cassettes inserted in 5′ and 3′ conserved regions using 3′CS and 5′CS-targeted primers [29] (Appendix A). The DNA of *C. jejuni* ATCC 33560 and sterile saline were included in all PCR assays as positive and negative controls, respectively.

### 2.5. Characterization of Gene Cassettes Arrays by DNA Sequencing

One of each amplified PCR product of repetitive distinct close size was selected, purified by PureLink PCR purification kit (Qiagen, Valencia, Spain) and sequenced using Big Dye Terminator V3.1 cycle sequencing kit (Perkin-Elmer Gmbh, Rodgau, Germany) in an Applied Biosystems 3130 genetic analyzer (California, USA). The resulting sequences were assembled using the SeqMan program within the Laser gene suite version 7 (DNAstar, Inc., Madison, WI, USA), then compared with the sequences in the GenBank database using the Basic Local Alignment Search Tool (http://www.ncbi.nlm.nih.gov/BLAST). The best BLAST hits on our query nucleotide sequences were selected based on the highest identity in the GenBank database. Alignment of the nucleotide sequences was performed using ClustalW sequence alignments (http://www.ebi.ac.uk/clustalw), then translation into amino acid sequences was performed using the ExPASy Translate Tool (http://us.expasy.org/, Swiss Institute of Bioinformatics SIB, Geneva, Switzerland). The novel complete gene cassette array (*aacC5-aadA7Δ4)* generated here assigned a new in number (in 1983) using the Integron Database INTEGRALL (http://integrall.bio.ua.pt/).

### 2.6. Bioinformatics and Statistical Analysis

Statistical Package for Social Sciences software (SPSS; v. 25, IBM, Armonk, NY, United States) was used for statistical analysis of data. Chi-squared test was used to determine if there were significant differences in the occurrence of antimicrobial resistance among different hosts (i.e., cattle, chicken, human) and between the two *Campylobacter* species being studied (*C. jejuni* and *C. coli*). *P* value was considered significant if <0.05. The overall distribution of the resistance phenotypes in *Campylobacter* isolates was visualized using a heat map. The clustering pattern of the isolates and the antimicrobial resistance phenotypes were determined by the hierarchical clustering dendrogram [30]. To predict the correlation among integron patterns and antimicrobial resistance phenotypes, correlation analyses were done on the raw data after conversion to a binary outcome (1 = variable presence, 0 = variable absence). The significance of the correlation was estimated at a significance level of 0.05. The variables ampicillin, amoxicillin, cephalothin, erythromycin, and sulfamethoxazole-trimethoprim were excluded from the analyses as they were identical among all isolates under study. The correlation analyses and visualization were done using R packages *corrplot*, *heatmaply*, *hmisc*, and *ggpubr* [31,32,33]. To estimate the similarities among *Campylobacter* isolates concerning various analyzed hosts (*n* = 3), the binary distances were calculated based on the presence or absence of certain integron patterns. This analysis was done using the functions *dist* and *hlcust* in the R environment.

### 2.7. Nucleotide Sequence Accession Numbers

DNA sequences generated in this study were submitted to GenBank and assigned the accession numbers of MT612446-MT612453.

## 3. Results

### 3.1. Prevalence of Campylobacter Species in Livestock Animals and Humans

As shown in Table 1, the overall occurrence rate of *Campylobacter* species was 42.72% (235/550), which significantly (*p* < 0.05) differed between species being 71.48% (168/235) for *C. jejuni* and 28.51% (67/235) for *C. coli*. Out of 265 samples of chicken origin, 154 (58.11%) *Campylobacter* isolates were detected, represented as 67.53% *C. jejuni* and 32.47% *C. coli*. The higher prevalence of *C. jejuni* was detected in chicken organs (61.54%), followed by cloacal swabs (57.14%) and chicken muscles (48.00%), while the isolation rate of *C. coli* from these sources was close to 30%, each. Moreover, *C. jejuni* were isolated from 18 of 35 (51.42%) minced meat samples, 26 of 35 (74.28%) raw milk, and 20 of 30 (66.67%) human stool samples, while *C. coli* were recorded by lower percentages. On the other hand, processed food products including chicken and meat luncheon, chicken and meat beef, smoked meat, canned milk, and canned and raw cheese were free from *Campylobacter* contamination. *Campylobacter* isolates yielded characteristic small, shiny, round, and gray colonies on mCCDA agar and no hemolysis on Columbia blood agar. All isolates were positive for oxidase, catalase, and nitrate reduction testing and exhibited sensitivity to nalidixic acid and resistance to cephalothin. *C. jejuni* isolates could hydrolyze indoxyl acetate and hippurate, while *C. coli* were indoxyl acetate-positive and hippurate-negative. *Campylobacter* isolates were further confirmed by PCR-based detection of the genus (*23S rRNA*) and species-specific (*mapA* for *C. jejuni* and *ceuE* for *C. coli)* genes.

### 3.2. Antimicrobial Resistance Profiles

The in vitro antimicrobial susceptibilities of 235 *Campylobacter* isolates comprising 168 *C. jejuni* and 67 *C. coli* against 25 antimicrobial agents are summarized in Table 2. The results revealed that all *Campylobacter* isolates originating from animal and human sources were resistant to amoxicillin, ampicillin, erythromycin, cephalothin, and sulfamethoxazole-trimethoprim (100%, each). Moreover, high levels of resistance were recorded for clarithromycin (100% and 97%), clindamycin (96.4% and 95.5%), nalidixic acid (90.5% and 86.6%), amoxycillin-clavulanic acid (89.3% and 80.6%), cefepime (88.1% and 83.6%), doxycycline (86.3% and 86.5%), colistin (83.9% and 88%) and chloramphenicol (83.3% and 80.6%) for *C. jejuni* and *C. coli* isolates, respectively. On the other hand, lower resistance rates were reported for amikacin (21.4% and 20.9%) and cefoxitin (26.8% and 43.2%) against *C. jejuni* and *C. coli* isolates, respectively. Of note, *C. jejuni* and *C. coli* were resistant to meropenem with alarming percentages (19.6% and 32.8%, respectively). Statistical analysis revealed significant differences in the resistance of *Campylobacter* species isolated from different sources to the most tested antimicrobials (*p* < 0.05) except for ampicillin-sulbactam that showed non-significant variation (*p* > 0.05). However, non-significant differences (*p* > 0.05) were reported between resistance of *C. jejuni* and *C. coli* to almost half of the examined antimicrobial agents.

As shown in Figure 1 and Appendix A, the antibiogram analysis revealed that *Campylobacter* isolates showed resistance to 11–25 antimicrobial agents with MAR indices ranged from 0.44 to 1.00 and demonstrated 93 distinct resistance patterns. The antibiotype 55 was the most prevalent among the analyzed isolates (*n* = 8; 3.40%) (Appendix A).

The PDR, XDR, and MDR patterns were reported among the analyzed isolates (Table 3 and Appendix A). In total, 2.55% (6/235) of *Campylobacter* isolates exhibited PDR patterns being resistant to all tested antimicrobial agents. The XDR profiles were extremely increased among analyzed isolates with a percentage of 68.94% (162/235). However, 28.5% (67/235) of the isolates showed MDR patterns. None of the examined *Campylobacter* isolates was pan-susceptible. Regarding the isolation source, most *C. jejuni* (68.26%) and *C. coli* (74%) isolates originated from chicken samples were XDR. All *C. jejuni* and *C. coli* isolates recovered from raw milk showed XDR and MDR patterns, respectively. Moreover, all *C. coli* isolated from human stool and minced meat exhibited MDR and XDR profiles, respectively, while 88.9% of *C. jejuni* originated from the minced meat were XDR. The PDR *Campylobacter* isolates originated from chicken cloacal swabs (*n* = 2) and human stool (*n* = 4).

### 3.3. Screening for Class 1 Integrons and Characterization of Associated Gene Cassettes in Campylobacter Isolates

Thirty-eight *Campylobacter* isolates (28 *C. jejuni*, and 10 *C. coli*) categorized as MDR (*n* = 5), XDR (*n* = 31), and PDR (*n* = 2) representing all sample origins and being resistant to at least 15 antimicrobial agents were screened for the possession of class 1 integrons using PCR assay. Overall, 37 of 38 (97.36%) examined isolates were positive for the *integrase* gene *(intI1)*, all harbored class 1 integrons carrying gene cassettes of varying sizes ranging from 349 to 2600 bp. Only one *C. jejuni* isolate (code No. 16) possessed an empty integron structure with no gene cassettes inserted between its conserved segments (Table 4). Eight repetitive distinct gene cassettes were selected among integron positive isolates for DNA sequencing. Other gene cassette arrays were identified according to their PCR product sizes based on relevant previously published data.

As shown in Table 4, 16 gene cassette arrays were identified among class1 integron-positive isolates. DNA sequence analysis revealed the predominance of *aadA* alleles (*aadA1a*, *aadA2*, *aadA5*, *aadA7Δ4,* and *aadA22*) in 30 out of 37 (81.08%) analyzed isolates, accounting resistance for aminoglycosides, particularly streptomycin. Other frequent gene cassettes reported herein were *dfrA* (25/37; 67.56%) alleles (*dfrA1*, *dfrA1*, *dfrA12*, *dfrA15,* and *dfrA17*), conferring resistance to the trimethoprim antimicrobial agent. Despite the high frequency of resistant *Campylobacter* isolates to B-lactams, the *bla pse-1* and *oxa1* gene cassettes were detected in only four isolates (4/37; 10.81% each). Likewise, the gene cassette *aacA4-cmlA4* conferring resistance to chloramphenicol was found in only two *Campylobacter* isolates.

The most striking finding in the current study is the exclusive existence of a novel gene cassette array namely *aacC5-aadA7Δ4* (In number in 1983) as a first report according to the INTEGRALL database in only two *Campylobacter* isolates (code Nos. 26 and 32). This conferred resistance to aminoglycosides in particular gentamicin and streptomycin, but not tobramycin, amikacin, nor kanamycin.

Two gene cassette arrays were reported within class 1 integrons of *Campylobacter* species, each one harbored triple genes. The *dfrA17-gcu5-aadA5* (1900 bp) integron-borne cassette array existed in two XDR (code Nos. 13 and 24) and one PDR (code No. 22) *Campylobacter* isolates and *dfrA12*-*gcu*-*aadA2* (1864 bp) gene cassette incorporated in four XDR *Campylobacter* isolates (code Nos. 4, 6, 12 and 17), both cassettes conferred resistance to aminoglycosides and trimethoprim antimicrobial agents (Table 4).

Of interest, all gene cassettes reported here were linked to antimicrobial resistance except one, whose product is a putative phage tail tape measure protein (349 bp; accession number MT612449). It was reported in three XDR *Campylobacter* isolates (code Nos. 7, 31, and 36) originated from chicken and one MDR *C. jejuni* isolate (code No. 14) of human origin, thus facilitates DNA transit to the cell cytoplasm during infection. To our knowledge, this is the first report of a putative phage tail protein associated with class 1 integrons in *Campylobacter* species.

### 3.4. Correlation between Class 1 Integrons and Antimicrobial Resistance Phenotypes in Campylobacter Isolates

As depicted in Figure 2 and Appendix A, PCR results and DNA sequence analysis were consistent with certain antimicrobial susceptibility phenotypes. It was noted that the existence of *aadA* and *aacC5* genes positively correlated (*r* = 0.09–0.18) with streptomycin resistance. Moreover, class 1 integron-positive isolates carrying *aacC5-aadA7Δ4* and *aacA4-cmlA4* cassette arrays showed positive correlations with resistance to gentamicin (*r* = 0.11) and tobramycin (*r* = 0.17), respectively. The presence of *bla pse-1* and *oxa1* genes non-significantly (*p* > 0.05) associated with resistance to amoxicillin-clavulanate (*r* = 0.12 each) and cefoperazone (*r* = 0.21 and 0.01, respectively). However, both genes did not confer resistance to ampicillin-sulbactam (*r* = −0.07 and −0.25, respectively), cefoxitin (*r* = −0.21 and −0.04, respectively) or cefepime (*r* = −0.16 each).

The clustering pattern of class 1 integron-positive *Campylobacter* isolates is illustrated in Figure 2. The two variables (gene cassette arrays and antimicrobial resistance phenotypes) produced two distinct clusters (A and B). Notably, the *aacA4-cmlA4* gene cassette gathered with tobramycin in cluster A. While, the *aadA* genes and *bla pse-1* and *oxa1-aadA1* cassette arrays, which confer resistance to streptomycin and amoxicillin-clavulanate and cefoperazone, respectively, were clustered together in cluster B.

### 3.5. Cluster Analysis of Gene Cassette Arrays in Campylobacter Isolates from Human and Animal Populations

The dendrogram analysis (Figure 3) of class 1 integron-positive isolates (*n* = 38) simplified the existence of gene cassettes across livestock animals and humans. Three clusters were noticed in our dataset (A, B and, C). A close relatedness was observed among certain *Campylobacter* isolates of different sources. As exemplified, a *Campylobacter* isolate of the chicken source (code CK153) was closer to another one of human origin (code H19), both were gathered in cluster A. In addition, two *Campylobacter* isolates of human (H21) and chicken (CK143) sources clustered closely together in cluster B. Regarding the cluster C, several isolates of the three populations (cattle, chicken and human) clustered together.

## 4. Discussion

Thermophilic campylobacters as *C. jejuni* and *C. coli* are associated with infections in humans due to the consumption of undercooked meat, particularly poultry, and unpasteurized milk [2,3]. Currently, increasing resistance to major antibiotics in use among campylobacters is an emerging problem [34]. This is the first report to provide insights into the carriage of class 1 integrons by PDR, XDR, and MDR *Campylobacter* species isolated from livestock animals and humans in Egypt. In this study, the overall occurrence rate of *C. jejuni* (71.48%) was higher than that of *C. coli* (28.51%). Most *Campylobacter* isolates were detected in chicken samples (58.11%) represented as 67.53% *C. jejuni* and 32.47% *C. coli*, while previous studies reported varying rates of *Campylobacter* prevalence in chickens ranging from 24% to 62% [35,36].

Raw milk acts as a second main source of campylobacters [37]. The consumption of unpasteurized milk and milk products has been implicated in infections of 23% of human cases with campylobacteriosis in Egypt [38]. We reported a *C. jejuni* prevalence rate of 74.29% in raw milk samples, which have been previously documented with a lower prevalence rate (34%) [39]. Cross-contamination with *Campylobacter* species could occur during slaughter and milking of cattle. Herein, the prevalence of *C. jejuni* in fresh meat was 51.43%, which was lower than that reported in a previous study in Ethiopia (72%) [40]. Of note, milk and meat products were free from *Campylobacter* species, which was consistent with a previous study in India [41].

Poultry, milk, and meat are a reservoir for campylobacters; therefore, food processing with poor sanitation is an important source of transmission leading to increase the risk of human exposure, especially those in contact with food-producing animals. A higher prevalence of *C. jejuni* in the human stool (66.67%) was detected in this study when compared with previous studies conducted in Egypt with prevalence rates of 27.5% [35], 16.66% [42], and 4.07% [43].

Campylobacteriosis does not usually require antibiotic treatment; however, in some cases, antibiotics may be administered. Ciprofloxacin and erythromycin are considered drugs of choice for treating *Campylobacter* infections in humans [44]. However, the unregulated use of antibiotics in human and veterinary medicine, resulting in increasing their resistance [45].

Increasing resistance to the major antibiotics in use among campylobacters is an emerging problem [34]. Therefore, an investigation of the resistance rates and mechanisms is essential to prevent the spread of antibiotic-resistant campylobacters in livestock animals and humans. Herein, we provided better insight into different drug resistance patterns (*n* = 93-pattern) as well as an alarming increase of PDR, XDR, and MDR categories, while testing 25 antimicrobials among 14 antimicrobial categories.

Resistance to three or more antimicrobial classes (MDR) is a worldwide disturbing situation in *C. jejuni* [46]. In this study, 28.5% of *Campylobacter* isolates originating from livestock animals and humans exhibited MDR pattern. This level of resistance was less than that reported previously [47], while the resistance profile reaching 11 to 19 drugs is worrying compared with previous studies that recorded resistance to five to six [48] or three to four antimicrobial agents [49].

Resistance to all classes of antimicrobial agents except two or fewer is defined as XDR [26]. Of interest, this is the first report of XDR *Campylobacter* isolates (68.94%) among livestock animals and humans. In a previous study, an XDR *C. jejuni* CCARM 13,322 was recovered from a human case of diarrhea associated with international travel [17]. Moreover, 2.55% of *Campylobacter* isolates showed PDR (resistant to all antimicrobial agents among 14 categories), which was not reported in any previous study yet.

Class 1 integron is the main cause of multiple antibiotic resistance gene cassettes transmission in Gram-negative bacteria causing multidrug resistance [9,50]. Till date, few reports detected class 1 integrons in both *C. jejuni* and *C. coli* isolated from chicken house environment [12] and human sources [13,14] without showing the antimicrobial resistance profiles of isolates. However, no reports could detect class 1 integrons in PDR or XDR *Campylobacter* isolates [17].

In this study, the interested report of *integrase* gene and class 1 integrons (97.36% each) in *C. jejuni* and *C. coli* of chicken, cattle, and human origins representing MDR, XDR, and PDR patterns were documented for the first time, at least in Egypt. These integrons were associated with gene cassettes of different sizes ranging from 349 to 2600 bp. Considering, the previously published data, all *Campylobacter* isolates originating from a chicken house environment harbored a single cassette in the integron with 900 bp amplicon [12]. Additionally, those isolates originating from human and poultry sources had gene cassettes of molecular weights ranging from 300 to 1.4 kb [14].

Class 1 integrons detected in this study associated with 16 resistance gene cassettes. The most frequently reported were trimethoprim (*dfrA1*, *dfrA1*, *dfrA12*, *dfrA15,* and *dfrA17*) and aminoglycoside (*aadA1a*, *aadA2*, *aadA5*, *aadA22,* and *aadA7Δ4*) resistance gene cassette arrays. Consistently, previous studies detected class 1 integrons associated with aminoglycoside resistance genes (*aadA2* and *aacA4*) in both *C. jejuni* and *C. coli* resulting from the sequencing of 1000 bp, and 900 bp amplicons, respectively [12,14]. Moreover, the trimethoprim resistance gene cassettes (*dfr1* and *dfr9*) were carried by class 1 integrons in clinical isolates of *C. jejuni* following the sequencing of 399 bp, and 254 bp amplicons, respectively [13,15].

The most surprising points in the current study are the carriage of a novel gene cassette array, namely *aacC5-aadA7Δ4* (in number in 1983) as a first report according to the INTEGRALL database. In addition, an unusual phage tail tape measure protein gene cassette was harbored by four *Campylobacter* isolates as a first record in campylobacters worldwide. It is widely assumed that *Campylobacter*–bacteriophage interactions may play a role in horizontal gene transfer. In accordance with previous reports, bacteriophages cause genomic instability in *C. jejuni* and mediate interstrain transfer of large DNA fragments [51,52].

Herein, the correlation between resistance to certain antimicrobials and the corresponding gene cassettes was shown similar to other studies as *Campylobacter* isolates harboring aminoglycoside resistance gene, *aacA4*, conferred higher tobramycin MICs but slightly increased resistance to gentamicin [12]. In addition, high-level resistance to trimethoprim in *C. jejuni* was associated with the acquisition of *dfr* genes [15].

Previous studies showed that the *dfr* cassette is mostly associated with the *aadA* gene cassette [53]. DNA sequence analysis of a 1513 bp amplicon of class 1 integron revealed *dfrA17ab*-*aadA5* gene cassette in seven XDR *Campylobacter* isolates. Moreover, *dfrA12*-*gcu*-*aadA2* gene cassette array was detected in four analyzed isolates. According to our results, the *dfrA1-aadA1* gene cassette has been reported in MDR *Citrobacter* species with 1600 bp fragment size [54]. In addition, a previous study detected these gene cassettes of 2000 bp in MDR *Salmonella* isolates [55].

## 5. Conclusions

This is the first report, at least in Egypt, that showed the prevalence of PDR and XDR *Campylobacter* species in livestock animals and humans. Moreover, we demonstrated the existence of class 1 integrons and associated gene cassettes in analyzed isolates, which confer antimicrobial resistance and the possibility of *Campylobacter*–bacteriophage interactions by the carriage of an unusual phage tail protein as a first report.

## Figures and Tables

**Figure 1 animals-10-02067-f001:**
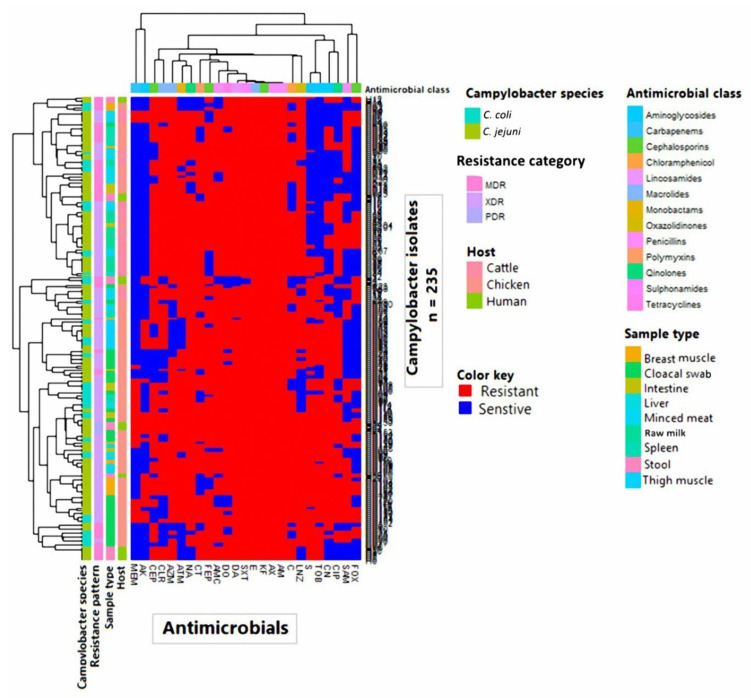
Overall distribution and clustering of *Campylobacter* isolates (*n* = 235) under study and the patterns of their antimicrobial resistance. Different *Campylobacter* species, hosts, sample types, antimicrobial classes, and resistance categories are shown for each isolate as color codes. The heat map represents the hierarchical clustering of the isolates and the antimicrobial classes.

**Figure 2 animals-10-02067-f002:**
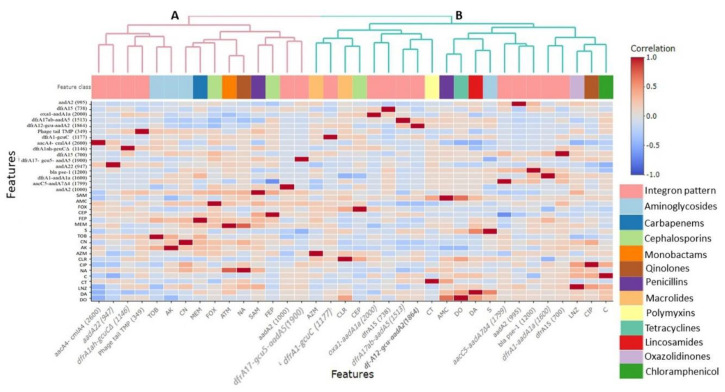
Pairwise correlation (*R*) among different antimicrobial resistance phenotypes and integron gene cassette arrays. Red and blue colors indicate positive and negative correlations, respectively. The color key refers to the correlation coefficient (*R*). The darker colors imply stronger positive or negative correlations. The hierarchical clustering of the variables is shown as a dendrogram illustrating different clusters with different colors and letters (e.g., A and B). Variables that are identical among all strains are excluded, and thus not shown in this figure. Classes of antimicrobials are color-coded below the dendrogram.

**Figure 3 animals-10-02067-f003:**
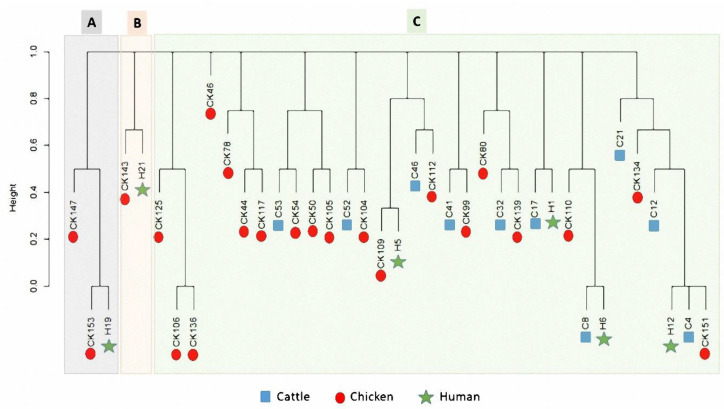
Dendrogram showing the binary distances among different *Campylobacter* isolates based on the integron patterns. The isolates are categorized based on their host, which are shown as different colors and symbols. The *X*-axis refers to the binary distance scale. Isolates codes are illustrated in Table 4.

**Table 1 animals-10-02067-t001:** Prevalence of *Campylobacter* species isolated from animal and human sources.

Source (No.)	Sample Type (No.)	Overall Prevalence of *Campylobacter* Isolates No. (%)	*Campylobacter* Species No. (%)		*p* Value
			*C. jejuni*	*C. coli*	
Chickens and chicken products (265)	Cloacal swab (70)	60 (85.71)	40 (57.14)	20 (28.57)	0.001
	Breast muscle (25)	18 (72.00)	14 (56.00)	4 (16.00)	0.003
	Thigh chicken muscle (25)	16 (64.00)	10 (40.00)	6 (24.00)	0.225
	Liver (25)	25 (100.00)	16 (64.00)	9 (36.00)	0.048
	Spleen (20)	18 (90.00)	14 (70.00)	4 (20.00)	0.001
	Intestine (20)	17 (85.00)	10 (50.00)	7 (35.00)	0.337
	Chicken beef (40)	0 (0.00)	0 (0.00)	0 (0.00)	NE
	Chicken luncheon (40)	0 (0.00)	0 (0.00)	0 (0.00)	NE
Meat and meat products (160)	Minced meat (35)	28 (80.00)	18 (51.43)	10 (28.57)	0.05
	Smoked meat (45)	0 (0.00)	0 (0.00)	0 (0.00)	NE
	Meat beef (40)	0 (0.00)	0 (0.00)	0 (0.00)	NE
	Meat luncheon (40)	0 (0.00)	0 (0.00)	0 (0.00)	NE
Dairy products (95)	Raw milk (35)	29 (82.86)	26 (74.29)	3 (8.57)	<0.0001
	Canned milk (20)	0 (0.00)	0 (0.00)	0 (0.00)	NE
	Canned cheese (20)	0 (0.00)	0 (0.00)	0 (0.00)	NE
	Raw cheese (20)	0 (0.00)	0 (0.00)	0 (0.00)	NE
Human (30)	Stool (30)	24 (80.00)	20 (66.67)	4 (13.33)	<0.0001
Total	550	235 (42.73)	168 (71.49)	67 (28.51)	<0.0001

NE, not estimated, *p* values < 0.05 are statistically significant.

**Table 2 animals-10-02067-t002:** Antimicrobial resistance pattern of *Campylobacter* species isolated from different sources.

AMA	Number of *C. jejuni* Isolates							Number of *C. coli* Isolates							*p* Value	
	Chickens (*n* = 104)			Raw Milk (26)	Minced Meat (18)	Human Stool (20)	Total (168)	Chickens (*n* = 50)			Raw Milk (3)	Minced Meat (10)	Human Stool (4)	Total (67)		
	Muscle (24)	Internal Organs (40)	Cloacal Swabs (40)					Muscle (10)	Internal Organs (20)	Cloacal Swabs (20)					Various Hosts	*Campylobacter* Species
AX	24	40	40	26	18	20	168	10	20	20	3	10	4	67	NA	NA
	(100.00)	(100.00)	(100.00)	(100.00)	(100.00)	(100.00)	(100.00)	(100.00)	(100.00)	(100.00)	(100.00)	(100.00)	(100.00)	(100.00)		
AM	24	40	40	26	18	20	168	10	20	20	3	10	4	67	NA	NA
	(100.00)	(100.00)	(100.00)	(100.00)	(100.00)	(100.00)	(100.00)	(100.00)	(100.00)	(100.00)	(100.00)	(100.00)	(100.00)	(100.00)		
SAM	15	27	20	11	10	10	93	0.00	11	16	3	1	0.00	31	0.1022	0.2
	(62.50)	(67.50)	(50.00)	(42.31)	(55.55)	(50.00)	(55.36)	(0.00)	(55.00)	(80.00)	(100.00)	(10.00)	(0.00)	(46.27)		
AMC	24	34	34	26	17	15	150	7	20	18	3	10	0.00	58	<0.0001	0.55
	(100.00)	(85.00)	(85.00)	(100.00)	(94.44)	(75.00)	(89.29)	(70.00)	(100.00)	(90.00)	(100.00)	(100.00)	(0.00)	(86.57)		
KF	24	40	40	26	18	20	168	10	20	20	3	10	4	67	NA	NA
	(100.00)	(100.00)	(100.00)	(100.00)	(100.00)	(100.00)	(100.00)	(100.00)	(100.00)	(100.00)	(100.00)	(100.00)	(100.00)	(100.00)		
FOX	8	4	18	17	12	9	68	0.00	4	10	0.00	0.00	4	18	0.002	0.05
	(33.33)	(10.00)	(45.00)	(65.38)	(66.67)	(45.00)	(40.48)	(0.00)	(20.00)	(50.00)	(0.00)	(0.00)	(100.00)	(26.87)		
CEP	14	34	19	26	12	18	123	10	20	12	3	9	4	58	0.0068	0.02
	(58.33)	(85.00)	(47.50)	(100.00)	(66.67)	(90.00)	(73.21)	(100.00)	(100.00)	(60.00)	(100.00)	(90.00)	(100.00)	(86.57)		
FEP	16	38	39	17	10	20	140	8	19	20	3	6	4	60	<0.0001	0.2
	(66.67)	(95.00)	(97.50)	(65.38)	(55.56)	(100.00)	(83.33)	(80.00)	(95.00)	(100.00)	(100.00)	(60.00)	(100.00)	(89.55)		
MEM	1	2	20	0.00	6	4	33	0.00	14	8	0.00	0.00	0.00	22	0.0124	0.03
	(4.17)	(5.00)	(50.00)	(0.00)	(33.33)	(20.00)	(19.64)	(0.00)	(70.00)	(40.00)	(0.00)	(0.00)	(0.00)	(32.84)		
ATM	14	39	36	26	17	10	142	0.00	14	12	3	10	4	43	<0.0001	0.0006
	(58.33)	(97.50)	(90.00)	(100.00)	(94.44)	(50.00)	(84.52)	(0.00)	(70.00)	(60.00)	(100.00)	(100.00)	(100.00)	(64.18)		
S	22	16	34	12	7	14	105	4	17	18	0.00	2	3	44	<0.0001	0.6
	(91.67)	(40.00)	(85.00)	(46.15)	(38.89)	(70.00)	(62.50)	(40.00)	(85.00)	(90.00)	(0.00)	(20.00)	(75.00)	(65.67)		
TOB	22	16	30	8	6	13	95	4	17	17	0.00	0.00	4	42	<0.0001	0.38
	(91.67)	(40.00)	(75.00)	(30.77)	(33.33)	(65.00)	(56.55)	(40.00)	(85.00)	(85.00)	(0.00)	(0.00)	(100.00)	(62.69)		
CN	24	24	36	14	2	9	109	6	17	14	0.00	5	4	46	<0.0001	0.58
	(100.00)	(60.00)	(90.00)	(53.85)	(11.11)	(45.00)	(64.88)	(60.00)	(85.00)	(70.00)	(0.00)	(50.00)	(100.00)	(68.66)		
AK	18	0.00	14	0.00	6	6	44	6	3	6	0.00	00.00	4	19	0.0032	0.73
	(75.00)	(0.00)	(35.00)	(0.00)	(33.33)	(30.00)	(26.19)	(60.00)	(15.00)	(30.00)	(0.00)	(0.00)	(100.00)	(28.36)		
E	24	40	40	26	18	20	168	10	20	20	3	10	4	67	NA	NA
	(100.00)	(100.00)	(100.00)	(100.00)	(100.00)	(100.00)	(100.00)	(100.00)	(100.00)	(100.00)	(100.00)	(100.00)	(100.00)	(100.00)		
AZM	15	33	32	26	18	20	144	4	20	14	3	10	4	55	<0.0001	0.48
	(62.50)	(82.50)	(80.00)	(100.00)	(100.00)	(100.00)	(85.71)	(40.00)	(100.00)	(70.00)	(100.00)	(100.00)	(100.00)	(82.09)		
CLR	24	40	6	26	18	20	134	10	20	2	3	10	0.00	45	<0.0001	0.04
	(100.00)	(100.00)	(15.00)	(100.00)	(100.00)	(100.00)	(79.76)	(100.00)	(100.00)	(10.00)	(100.00)	(100.00)	(0.00)	(67.16)		
CIP	24	16	32	26	1	11	110	6	9	16	0.00	9	0.00	40	<0.0001	0.4
	(100.00)	(40.00)	(80.00)	(100.00)	(5.56)	(55.00)	(65.48)	(60.00)	(45.00)	(80.00)	(0.00)	(90.00)	(0.00)	(59.70)		
NA	24	33	40	26	17	10	150	6	20	18	0.00	10	4	58	<0.0001	0.55
	(100.00)	(82.50)	(100.00)	(100.00)	(94.44)	(50.00)	(89.29)	(60.00)	(100.00)	(90.00)	(0.00)	(100.00)	(100.00)	(86.57)		
SXT	24	40	40	26	18	20	168	10	20	20	3	10	4	67	<0.0001	0.005
	(100.00)	(100.00)	(100.00)	(100.00)	(100.00)	(100.00)	(100.00)	(100.00)	(100.00)	(100.00)	(100.00)	(100.00)	(100.00)	(100.00)		
C	24	26	32	26	17	16	141	6	16	16	3	5	0.00	46	0.0461	0.0088
	(100.00)	(65.00)	(80.00)	(100.00)	(94.44)	(80.00)	(83.93)	(60.00)	(80.00)	(80.00)	(100.00)	(50.00)	(0.00)	(68.66)		
CT	14	28	34	26	18	11	131	6	16	18	3	10	4	57	<0.0001	0.21
	(58.33)	(70.00)	(85.00)	(100.00)	(100.00)	(55.00)	(77.98)	(60.00)	(80.00)	(90.00)	(100.00)	(100.00)	(100.00)	(85.07)		
LNZ	24	39	28	26	14	13	144	10	9	18	0.00	10	0.00	47	0.0012	0.005
	(100.00)	(97.50)	(70.00)	(100.00)	(77.78)	(65.00)	(85.71)	(100.00)	(45.00)	(90.00)	(0.00)	(100.00)	(0.00)	(70.15)		
DA	23	40	39	26	18	20	166	10	20	20	3	10	0.00	63	<0.0001	0.03
	(95.83)	(100.00)	(97.50)	(100.00)	(100.00)	(100.00)	(98.81)	(100.00)	(100.00)	(100.00)	(100.00)	(100.00)	(0.00)	(94.03)		
DO	24	37	24	26	18	20	149	10	20	18	3	10	0.00	61	0.0103	0.59
	(100.00)	(92.50)	(60.00)	(100.00)	(100.00)	(100.00)	(88.69)	(100.00)	(100.00)	(90.00)	(100.00)	(100.00)	(0.00)	(91.04)		

Values represent number of *Campylobacter* isolates (%), *p* values were calculated based on Chi-squared test; *p* values < 0.05 are statistically significant; *p* values < 0.01 are highly significant; AMA, antimicrobial agent; AX, amoxicillin; AM, ampicillin; SAM, ampicillin-sulbactam; AMC, amoxycillin-clavulanic acid; KF, cephalothin; FOX, cefoxitin; CEP, cefoperazone; FEP, cefepime; MEM, meropenem; ATM, aztreonam; S, streptomycin; TOB, tobramycin; CN, gentamicin; AK, amikacin; E, erythromycin; AZM, azithromycin; CLR, clarithromycin; CIP, ciprofloxacin; NA, nalidixic acid; SXT, sulfamethoxazole-trimethoprim; C, chloramphenicol; CT, colistin; LNZ, lenzolid; DA, clindamycin; DO, doxycycline; NA, non-applicable.

**Table 3 animals-10-02067-t003:** Occurrence of MDR, XDR and PDR categories in *Campylobacter* isolates from different sources.

Resistance Category	Resistance to Antimicrobial Class (*n* = 14)	Resistance to Antimicrobial Agent (*n* = 25)	No. of Resistant *Campylobacter* Isolates (Source)	
			*C. jejuni* (*n* = 168)	*C. coli* (*n* = 67)
MDR (*n* = 67)	7	11	0	2 (chicken muscle)
		12	0	1 (chicken muscle)
		14	0	1 (human stool)
		15	0	3 (human stool)
	8	13	0	1 (chicken muscle)
	10	13	2 (cloacal swab)	0
		14	1 (minced meat)	0
		15	8 (chicken organ)	0
		16	2 (cloacal swab), 9 (human stool)	3 (raw milk)
		17	5 (human stool), 1 (cloacal swab)	2 (cloacal swab)
	11	15	1 (minced meat)	2 (cloacal swab)
		16	4 (cloacal swab), 2 (chicken organ)	0
		17	8 (chicken organ), 2 (cloacal swab)	3 (chicken organ)
		18	1 (cloacal swab)	0
		19	1 (chicken muscle)	2 (cloacal swab)
XDR (*n* = 162)	12	16	2 (cloacal swab)	0
		17	3 (minced meat)	4 (minced meat)
		18	2 (chicken muscle), 5 (chicken organ), 4 (cloacal swab)	4 (minced meat), 2 (chicken muscle), 2 (cloacal swab)
		19	5 (chicken muscle), 5 (chicken organ), 2 (cloacal swab), 4 (minced meat)	1 (minced meat), 1 (chicken organ), 2 (cloacal swab)
		20	5 (raw milk), 2 (chicken muscle), 2 (cloacal swab), 2 (chicken organ), 2 (human stool)	6 (chicken organ), 4 (chicken muscle)
		21	2 (chicken muscle)	4 (chicken organ)
		22	4 (chicken muscle)	0
		23	3 (chicken muscle)	0
	13	18	1 (minced meat)	1 (minced meat), 2 (cloacal swab)
		19	9 (raw milk), 1 (chicken muscle), 3 (chicken organ)	1 (cloacal swab)
		20	2 (minced meat), 5 (raw milk)	1 (cloacal swab)
		21	6 (raw milk), 3 (chicken muscle), 6 (cloacal swab)	3 (chicken organ)
		22	1 (raw milk), 5 (chicken organ), 2 (cloacal swab)	0
		23	1 (chicken muscle)	0
	14	18	1 (cloacal swab)	0
		19	1 (cloacal swab)	0
		20	6 (minced meat)	0
		21	1 (cloacal swab)	0
		22	3 (cloacal swab)	0
		23	2 (cloacal organ)	3 (chicken organ), 4 (cloacal swab)
		24	2 (cloacal swab)	2 (cloacal swab)
PDR (*n* = 6)	14	25	4 (human stool), 2 (cloacal swab)	0

MDR, multiple drug-resistance; XDR, extensively drug-resistance; PDR, pan drug-resistance.

**Table 4 animals-10-02067-t004:** Antimicrobial resistance patterns and gene cassette arrays carried by class 1 integrons in MDR, XDR and PDR *Campylobacter* isolates recovered from different sources (*n* = 38).

Isolate No.	Code No.	*Campylobacter* Species	Source	Antimicrobial Resistance Pattern	Resistance to Antimicrobial Agents	Resistance to Antimicrobial Classes	Antimicrobial Resistance Type	intI1/Class 1 Integron	Resistance Gene(s) in Class 1 Integron (Size in bp)
1	CK105	*C. coli*	Chicken intestine	AX, AM, SAM, AMC, KF, FOX, CEP, FEP, MEM, ATM, S, TOB, CN, E, AZM, CLR, NA, SXT, CT, DA, DO	21	12	XDR	+/+	*aadA2* (995)
2	CK134	*C. jejuni*	Chicken cloacal swab	AX, AM, SAM, AMC, KF, FOX, FEP, MEM, ATM, TOB, CN, E, AZM, CIP, NA, SXT, C, CT, LNZ, DA, DO	21	14	XDR	+/+	*dfrA15* (738)
3	CK110	*C. jejuni*	Chicken thigh muscle	AX, AM, SAM, AMC, KF, FEP, ATM, S, TOB, CN, E, AZM, CLR, CIP, NA, SXT, C, CT, LNZ, DA, DO	21	13	XDR	+/+	*dfrA15* (738); *oxa1-aadA1a* (2000)
4	H12	*C. jejuni*	Human stool	AX, AM, KF, FOX, CEP, FEP, S, CN, E, AZM, CLR, CIP, SXT, C, LNZ, DA, DO	17	10	MDR	+/+	*dfrA17ab-aadA5* (1513); *dfrA12-gcu-aadA2* (1864)
5	H19	*C. jejuni*	Human stool	AX, AM, SAM, KF, FOX, FEP, ATM, S, TOB, CN, AK, E, AZM, CLR, NA, SXT, C, LNZ, DA, DO	20	12	XDR	+/+	*dfrA15* (738); *dfrA17ab-aadA5* (1513); *oxa1-aadA1a* (2000)
6	CK139	*C. jejuni*	Chicken breast muscle	AX,AM, SAM, AMC, KF, FOX, CEP, FEP, MEM, ATM, S, CN, TOB, E, AZM, CLR, CIP, NA, SXT, C, LNZ, DA, DO	23	13	XDR	+/+	*dfrA12-gcu-aadA2* (1864)
7	CK104	*C. jejuni*	Chicken intestine	AX, AM, SAM, AMC, KF, FOX, CEP, FEP, ATM, E, AZM, CLR, CIP, NA, SXT, C, CT, LNZ, DA, DO	20	12	XDR	+/+	Phage tail TMP (349)
8	CK147	*C. jejuni*	Chicken Liver	AX, AM, AMC, SAM, KF, FEP, CEP, FOX, MEM, ATM, S, TOB, CN, E, AZM, CLR, NA, SXT, C, CT, LNZ, DA, DO	23	14	XDR	+/+	*dfrA1-gcuC* (1177)
9	C41	*C. jejuni*	Raw milk	AX, AM, SAM, AMC, KF, FOX, CEP, FEP, ATM, CN, E, AZM, CLR, CIP, NA, SXT, C, CT, LNZ, DA, DO	21	13	XDR	+/+	*aacA4-cmlA4* (2600)
10	CK143	*C. coli*	Chicken cloacal swab	AX, AM, SAM, AMC, KF, FOX, FEP, MEM, ATM, S, TOB, CN, AK, E, AZM, CIP, NA, SXT, C, CT, LNZ, DA, DO	23	14	XDR	+/+	*dfrA15* (738); *dfrA17ab-aadA5* (1513)
11	H1	*C. coli*	Human stool	AX, AM, KF, FOX, CEP, FEP, ATM, TOB, CN, AK, E, AZM, NA, SXT, CT	15	8	MDR	+/+	***dfrA15* (700); *dfrA1ah-gcuCΔ* (1146)**
12	C52	*C. jejuni*	Raw milk	AX, AM, SAM, AMC, KF, FOX, CEP, FEP, ATM, S, CN, E, AZM, CLR, CIP, NA, SXT, C, CT, LNZ, DA, DO	22	13	XDR	+/+	*dfrA17ab-aadA5* (1513); *dfrA12-gcu-aadA2* (1864)
13	C8	*C. jejuni*	Minced meat	AX, AM, AMC, SAM, KF, FEP, ATM, S, E, AZM, CLR, NA, SXT, C, CT, LNZ, DA, DO	18	12	XDR	+/+	*dfrA17-gcu5-aadA5* (1900)
14	H5	*C. jejuni*	Human stool	AX, AM, AMC, KF, CEP, FEP, S, TOB, E, AZM, CLR, SXT, C, CT, DA, DO	15	10	MDR	+/+	Phage tail TMP (349)
15	CK54	*C. coli*	Chicken cloacal swab	AX, AM, AMC, KF, FEP, ATM, S, TOB, CN, E, CIP, NA, SXT, C, CT, LNZ, DA, DO	18	13	XDR	+/+	*aadA22* (947); *dfrA17ab-aadA5* (1513)
16	CK80	*C. jejuni*	Chicken breast muscle	AX, AM, SAM, AMC, KF, CEP, FEP, TOB, CN, E, AZM, CLR, CIP, NA, SXT, C, CT, LNZ, DO	19	11	MDR	-/-	ND
17	CK125	*C. jejuni*	Chicken Liver	AX, AM, SAM, AMC, KF, FEP, CEP, ATM, S, TOB, CN, E, AZM, CLR, CIP, NA, SXT, C, CT, LNZ, DA, DO	22	13	XDR	+/+	*dfrA1-gcuC* (1177); *dfrA12-gcu-aadA2* (1864)
18	C12	*C. coli*	Minced meat	AX, AM, AMC, KF, CEP, FEP, ATM, S, CN, E, AZM, CLR, CIP, NA, SXT, CT, LNZ, DA, DO	19	12	XDR	+/+	*dfrA15* (700); *dfrA1-gcuC* (1177); *dfrA17ab-aadA5* (1513)
19	CK78	*C. jejuni*	Chicken cloacal swab	AX, AM, AMC, KF, FEP, MEM, ATM, S, CN, E, AZM, CIP, NA, SXT, C, CT, LNZ, DA, DO	19	14	XDR	+/+	*dfrA15* (738); *aadA22* (947); *dfrA1ah-gcuCΔ* (1146)
20	C46	*C. jejuni*	Raw milk	AX, AM, AMC, KF, FOX, CEP, FEP, ATM, S, CN, E, AZM, CLR, CIP, NA, SXT, C, CT, LNZ, DA, DO	21	13	XDR	+/+	*aadA2* (995); *bla pse-1* (1200)
21	CK109	*C. jejuni*	Chicken breast muscle	AX, AM, SAM, AMC, KF, CEP, FEP, ATM, S, CN, E, AZM, CLR, CIP, NA, SXT, C, CT, LNZ, DA, DO	21	13	XDR	+/+	*dfrA15* (700); *dfrA1-aadA1a* (1600)
22	CK153	*C. jejuni*	Chicken cloacal swab	AX, AM, SAM, AMC, KF, FOX, CEP, FEP, MEM, ATM, S, TOB, CN, AK, E, AZM, CLR, CIP, NA, SXT, C, CT, LNZ, DA, DO	25	14	PDR	+/+	*dfrA15* (738); *dfrA17-gcu5-aadA5* (1900)
23	CK50	*C. jejuni*	Chicken liver	AX, AM, SAM, AMC, KF, FEP, CEP, ATM, S, TOB, E, CLR, NA, SXT, CT, LNZ, DA, DO	18	12	XDR	+/+	*dfrA15* (738); *bla pse-1* (1200); *dfrA1-aadA1a* (1600)
24	CK117	*C. jejuni*	Chicken cloacal swab	AX, AM, SAM, KF, CEP, FEP, MEM, ATM, S, TOB, CN, AK, E, AZM, CIP, NA, SXT, C, CT, LNZ, DA	21	13	XDR	+/+	*aadA2* (995); *dfrA17-gcu5-aadA5* (1900)
25	H21	*C. jejuni*	Human stool	AX, AM, SAM, AMC, KF, FOX, CEP, FEP, MEM, ATM, S, TOB, CN, AK, E, AZM, CLR, CIP, NA, SXT, C, CT, LNZ, DA, DO	25	14	PDR	+/+	*aadA22* (947); *aacA4-cmlA4* (2600)
26	C32	*C. jejuni*	Raw milk	AX, AM, AMC, KF, CEP, ATM, S, CN, E, AZM, CLR, CIP, NA, SXT, C, CT, LNZ, DA, DO	19	13	XDR	+/+	*dfrA15* (700*)*; *aadA2* (995); *bla pse-1* (1200); ***aacC5-aadA7Δ4* (1799)**
27	CK44	*C. jejuni*	Chicken thigh muscle	AX, AM, AMC, KF, S, TOB, CN, AK, E, CLR, CIP, NA, SXT, C, CT, LNZ, DA, DO	18	12	XDR	+/+	*aadA2* (1000)
28	H6	*C. jejuni*	Human stool	AX, AM, AMC, KF,CEP, FEP, S, TOB, E, AZM, CLR, SXT, C, CT, DA, DO	16	10	MDR	+/+	*aadA2* (995)
29	CK112	*C. coli*	Chicken Liver	AX, AM, AMC, KF, FEP, CEP, MEM, ATM, S, TOB, CN, E, AZM, CLR, CIP, NA, SXT, C, CT, DA, DO	21	13	XDR	+/+	***dfrA1-gcuC* (1177)**
30	C17	*C. jejuni*	Minced meat	AX, AM, AMC, KF, FOX, CEP, FEP, ATM, S, TOB, E, AZM, CLR, NA, SXT, C, CT, DA, DO	19	12	XDR	+/+	***dfrA15* (738)**; *aadA2* (1000)
31	CK106	*C. coli*	Chicken intestine	AX, AM, SAM, AMC, KF, FOX, CEP, FEP, MEM, ATM, S, TOB, CN, E, AZM, CLR, NA, SXT, CT, DA, DO	21	12	XDR	+/+	**Phage tail TMP (349); *aadA22* (947)**
32	CK99	*C. coli*	Chicken cloacal swab	AX, AM, SAM, AMC, KF, FOX, CEP, FEP, ATM, S, CN, E, NA, SXT, C, CT, LNZ, DA, DO	19	13	XDR	+/+	*dfrA15* (738); *aacC5-aadA7Δ4* (1799)
33	C53	*C. jejuni*	Raw milk	AX, AM, SAM, AMC, KF, FOX, CEP, FEP, ATM, S, CN, E, AZM, CLR, CIP, NA, SXT, C, CT, LNZ, DA, DO	22	13	XDR	+/+	*aadA2* (995)
34	C4	*C. coli*	Minced meat	AX, AM, AMC, KF, CEP, ATM, S, E, AZM, CLR, CIP, NA, SXT, C, CT, LNZ, DA, DO	18	13	XDR	+/+	*aadA22* (947)
35	C21	*C. jejuni*	Minced meat	AX, AM, AMC, KF, FOX, CEP, ATM, S, TOB, CN, E, AZM, CLR, NA, SXT, C, CT, LNZ, DA, DO	20	13	XDR	+/+	*oxa1-aadA1a* (2000)
36	CK46	*C. jejuni*	Chicken cloacal swab	AX, AM, AMC, KF, FEP, ATM, S, TOB, CN, E, AZM, CIP, NA, SXT, C, CT, DA, DO	18	12	XDR	+/+	Phage tail TMP (349); *dfrA1ah-gcuCΔ* (1146); *oxa1-aadA1a* (2000)
37	CK151	*C. coli*	Chicken cloacal swab	AX, AM, SAM, AMC, KF, FOX, FEP, MEM, ATM, S, TOB, CN, AK, E, AZM, CLR, CIP, NA, SXT, C, CT, LNZ, DA, DO	24	14	XDR	+/+	*dfrA15* (738); *aadA22* (947); *bla pse-1* (1200)
38	CK136	*C. jejuni*	Chicken breast muscle	AX, AM, SAM, AMC, KF, FOX, CEP, FEP, ATM, S, CN, TOB, AK, E, AZM, CLR, CIP, NA, SXT, C, LNZ, DA, DO	23	12	XDR	+/+	***dfrA17ab-aadA5* (1513)**

AX, amoxicillin; AM, ampicillin; SAM, ampicillin-sulbactam; AMC, amoxycillin-clavulanic acid; KF, cephalothin; FOX, cefoxitin; CEP, cefoperazone; FEP, cefepime; MEM, meropenem; ATM, aztreonam; S, Streptomycin; TOB, tobramycin; CN, gentamicin; AK, amikacin; E, erythromycin; AZM, azithromycin; CLR, clarithromycin; NA, nalidixic acid; CIP, ciprofloxacin; SXT, sulfamethoxazole-trimethoprim; C, chloramphenicol; CT, colistin; LNZ, lenzolid; DA, clindamycin; DO, doxycycline; MDR, multiple drug-resistance; XDR, extensively drug-resistance; PDR, pan drug-resistance; TMP, tape measure protein; ND, not detected; CK, chicken; H, human; C, cattle; +, positive; -, negative. Bold gene cassette arrays were subjected to DNA sequencing, deposited in the GenBank database and assigned their accession numbers as depicted in the Material and Methods section.

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
