# Peer review of "First Report of aacC5-aadA7Δ4 Gene Cassette Array and Phage Tail Tape Measure Protein on Class 1 Integrons of Campylobacter Species Isolated from Animal and Human Sources in Egypt"

_animals, 2020, doi:10.3390/ani10112067_

Round 1

Reviewer 1 Report

This is a thorough investigation into Campylobacter presence in multiple foodstuffs and patients. The data is detailed and the figures of high quality.

The correction points are minor:

The title should be altered to include the isolate collection and survey as well as the integrons.

Line 16 and elsewhere: it seems that none of the isolates were pan sensitive. If this is the case, this should be mentioned.

Line 67: a couple more references on Integrons in Campy could be included here.

Were the stool sample from patients suffering diarrhoea? This is not clear in the methods. 

Line 126: in which way did the genomic DNA exhibit AMR? Please alter.

Line 135: What is "distinctive close size"?

Line 247: "were existed"? Please alter.

Section 3.3: were all thes products run through Blast? What % Ids were retrieved? How is the novel gene cassette defined?

Section 3.5: I do not understand this dendrogram analysis: what data is input? I also do not see the human isolates clustering separately. This section should be revised or removed.

Discussion first paragraph: careful with use of majority (x2). These sentences are misleading.

Line 341: wrong use of concurrent.

Throughout: please replace "exaggerated" with a more appropraite word.

Throughout: please italicise and capitalise Campylobacter. Except in italicised titles, eg line 309, where it should be normal font (capitalised).

Author Response

Reviewer 1

This is a thorough investigation into Campylobacter presence in multiple foodstuffs and patients. The data is detailed and the figures of high quality.
The correction points are minor.

Author response: Many thanks for your careful revision. All your suggestions were addressed thoroughly in the revised manuscript and in the responses below.

Reviewer 1: The title should be altered to include the isolate collection and survey as well as the integrons.

Author response: Thank you for your thorough comments. The title was changed as your recommendation.

Reviewer 1: Line 16 and elsewhere: it seems that none of the isolates were pan sensitive. If this is the case, this should be mentioned.

Author response: Thank you for your comment. This was mentioned in the revised version of the manuscript (Lines 17-18; lines 246-247).

Reviewer 1: Line 67: a couple more references on Integrons in Campy could be included here.

Author response: Thank you for your insightful comment. More references regarding Campylobacter harboring class 1 integrons were provided in the revised manuscript (please show the references No. 15 and 16).

Reviewer 1: Were the stool sample from patients suffering diarrhea? This is not clear in the methods. 

Author response: Yes, human stool samples were obtained from patients suffered from diarrhea. This was clarified in the materials and methods section (Lines 80, 81).

Reviewer 1: Line 126: in which way did the genomic DNA exhibit AMR? Please alter.
Author response: Thank you for your thorough comment. It was corrected in the revised manuscript (Lines 128-130).

Reviewer 1: Line 135: What is "distinctive close size"?

Author response: Thank you for your question. It was corrected to “repetitive distinct close size”, this mean we selected the repeated distinct gene cassettes among integron positive isolates for DNA sequencing. One only of the gene cassettes of nearly similar sizes was exposed to DNA sequencing.

Reviewer 1: Line 247: "were existed"? Please alter.

Author response: Thank you for your comment. It was corrected in the revised manuscript.

Reviewer 1: Section 3.3: were all these products run through Blast? What % Ids were retrieved? How is the novel gene cassette defined?

Author response:

  • Yes, the BLAST identity percentages are ranged from 98.69 -100% for all class 1 integron gene cassette arrays except the newyly reported one. Also, tap measured protein reported here was identical to that previously reported in an coli strain with an identity % of 100%.
  • The novel complete gene cassette array (aacC5- aadA7Δ4) in only two campylobacter isolates (code Nos. 26 and 32) generated herein assigned a new In number (In 1983) through the Integron Database INTEGRALL (http://integrall.bio.ua.pt/).

Reviewer 1: Section 3.5: I do not understand this dendrogram analysis: what data is input? I also do not see the human isolates clustering separately. This section should be revised or removed.

Author response: Thank you for your comment.  The input data is the identified gene cassette arrays on class 1 integrons of Campylobacter isolates of human and livestock animals (chickens and cows). This section was revised and corrected in the revised manuscript (Lines 342-349).

Reviewer 1: Discussion first paragraph: careful with use of majority (x2). These sentences are misleading.

Author response: Thank you; it was corrected in the revised version of the manuscript.

Reviewer 1: Line 341: wrong use of concurrent.

Author response: Thank you; it was corrected in the revised version of the manuscript.

Reviewer 1: Throughout: please replace "exaggerated" with a more appropriate word.

Author response: Thank you; it was changed.

Reviewer 1: Throughout: please italicise and capitalise Campylobacter. Except in italicised titles, eg line 309, where it should be normal font (capitalised).

Author response: Thank you; it was corrected in the revised version of the manuscript.

Reviewer 2 Report

The article ID animals-971009 investigated for the first time, the class I integrons and associated gene cassettes among pan drug resistant, extensively drug resistant and multidrug resistant Campylobacter species isolated from livestock animals and humans in Egypt, providing new insights into campylobacter antibiotic-resistance mechanisms. Manuscript is well written and the methods and results are well described. However, some minor issues should be addressed.

Specifically, the authors should check references carefully e.g. reference n. 20 does not match with the guidelines of Clinical and Laboratory Standards Institute (CLSI).

Strength of this study: This is the first report of aacC5-aadA7Δ4 gene cassette array and a putative phage tail protein on class 1 integrons of Campylobacter species. Weakness of this study: Although the study is well explained, there are too many tables and figures which can make reading difficult. Moreover, I suggest the authors italicize "Campylobacter" throughout the text.

Author Response

Reviewer 2:

The article ID animals-971009 investigated for the first time, the class I integrons and associated gene cassettes among pan drug resistant, extensively drug resistant and multidrug resistant Campylobacter species isolated from livestock animals and humans in Egypt, providing new insights into campylobacter antibiotic-resistance mechanisms. Manuscript is well written and the methods and results are well described. However, some minor issues should be addressed.

Reviewer 2: Specifically, the authors should check references carefully e.g. reference n. 20 does not match with the guidelines of Clinical and Laboratory Standards Institute (CLSI).

Author response: Thank you for your comment. The references were checked carefully and corrected.

Reviewer 2: Strength of this study: This is the first report of aacC5-aadA7Δ4 gene cassette array and a putative phage tail protein on class 1 integrons of Campylobacter species. Weakness of this study: Although the study is well explained, there are too many tables and figures which can make reading difficult. Moreover, I suggest the authors italicize "Campylobacter" throughout the text.

Author response:

  • Due to the large no of examined samples as well as the large No. of Campylobacter isolates, I think these Tables and Figures are helpful for the readers. Also, many of these Tables and Figures present as supplementary files.
  • Campylobacter was italicize throughout the text of the revised manuscript.
